# Biotic Environments Supporting the Persistence of Clinically Relevant Mucormycetes

**DOI:** 10.3390/jof6010004

**Published:** 2019-12-20

**Authors:** Malcolm D. Richardson, Riina Rautemaa-Richardson

**Affiliations:** 1Mycology Reference Centre Manchester, ECMM Excellence Centre, Manchester University NHS Foundation Trust, Wythenshawe Hospital, Manchester M23 9LT, UK; riina.richardson@manchester.ac.uk; 2Division of Infection, Immunity & Respiratory Medicine, School of Biological Sciences, Faculty of Biology, Medicine and Health, The University of Manchester, Manchester M13 9NT, UK; 3Department of Infectious Diseases, Manchester University NHS Foundation Trust, Wythenshawe Hospital, Manchester M23 9LT, UK

**Keywords:** Mucorales, mucormycosis, ecological niches, spore dispersal

## Abstract

Clinically relevant members of the Mucorales group can grow and are found in diverse ecological spaces such as soil, dust, water, decomposing vegetation, on and in food, and in hospital environments but are poorly represented in mycobiome studies of outdoor and indoor air. Occasionally, Mucorales are found in water-damaged buildings. This mini review examines a number of specialised biotic environments, including those revealed by natural disasters and theatres of war, that support the growth and persistence of these fungi. However, we are no further forward in understanding exposure pathways or the chronicity of exposure that results in the spectrum of clinical presentations of mucormycosis.

## 1. Introduction

The natural habitats of mucoraceous moulds have been exhaustively reviewed up to 2000 by Ribes and colleagues [1]. It is clear that there is a vast diversity of environments where these fungi have been found. However, it is not clear whether the growth of Mucorales in these habitats constitutes a major exposure and infection risk. A further review examined some of these niches to suggest ways in which vulnerable patients are exposed to these opportunistic fungi [2]. There are three basic exposure pathways: inhalation, ingestion, or direct contact. The degree or extent of exposure is determined by measuring the amount of the hazardous substance at the point of contact. With regard to exposure to Mucorales propagules (sporangiospores), it is difficult to determine an exact time of contact or over how long a time period chronic exposure had taken place.

The present brief review focusses on what has been learnt about the ecology of the Mucorales in the past few years. Many authoritative texts state that these fungi are ubiquitous with worldwide distribution, are thermotolerant and grow on decaying organic matter. Here, we examine these presumptions.

Most species of the Mucorales group are saprobic, though some attack other fungi as well as animals and plants. For example, some species of *Rhizopus* and *Lichtheimia* are widely distributed on stored grain, fruit and vegetables, in the air or in compost, as well as a number of circumscribed indoor environments. A number of species cause rot in ripe and harvested fruit and vegetables, and grow on animal excrement. All of these local environmental niches suggest exposure pathways for patients at risk. In addition to these well-described habitats, Mucorales appear to persist in a variety of unusual environmental spaces which are reviewed here.

Despite the increase in the number of human cases reported over the past decade, the more recent environmental microbiology literature provides limited insights into how common Mucorales are in the environment, and provides few clues about which ecological niches these fungi are found in. Examination of air sampling surveys in indoor and outdoor environments could indicate the level of exposure or help explain the apparent seasonality of mucormycosis. Similar analyses of other environments might reveal specific point sources of fungal communities. Of particular interest is how natural disasters and well described cases of cutaneous mucormycosis have revealed new, previously unknown fungal habitats. An understanding of exposure pathways in the acquisition of mucormycosis is based on what is known about the biology of the causative fungi, their natural and not so natural habitats and mechanisms of spore dispersal. These topics are discussed here. An up-to-date examination of the molecular taxonomy of the pathogenic Mucorales can be found in [3] and other articles in the special issue of the *Journal of Fungi* devoted to Mucorales and mucormycosis [4].

## 2. Optimal Environmental Conditions for Growth and Sporulation

An understanding of the environmental and nutritional requirements of Mucorales will help to indicate on which substrates these fungi may be found and an insight into exposure by vulnerable hosts [5]. Mucorales have a wide tolerance of temperatures and depending on the temperature range for growth, they can be allocated to different classes [5]. The highest tolerance is found in the psychrophiles and the thermophiles. A good example of a mucoraceous thermophile is *Rhizomucor pusillus* with a maximum growth temperature of 54–58 °C. Other examples of thermophilic Mucorales include *Lichtheimia corymbifera* which can grow up to 45–50 °C. *Apophysomyces elegans* grows at temperatures up to 42 °C. *Rhizopus* species grow well at 37 °C.

Irrespective of the status of any material, whether a fungus grows on and in that material or not depends on the availability of water in the material. Individual fungi have their own particular moisture requirements. Growth is dependent on enough free water being available. This water does not include bound water. The term water availability (a_w_) is used to denote the amount of available water in a substrate. Moulds have a minimum and an optimum moisture requirement for growth. Most optima are in the range of 0.90–0.99. The few species of Mucorales where the a_w_ has been determined fall into a group of filamentous fungi classified as slightly xerophilic (minimum a_w_ 0.80–0.89), for example *Lichtheimia corymbifera* [6].

Glucose is the main carbon source for growth for Mucorales [5]. Like other fungi, Mucorales have a requirement for proteins and a range of organic and inorganic substances. The pH of a particular environment can markedly affect the rate of hyphal extension, growth, and sporulation. Like temperature, it can markedly affect the rate of extension growth [5]. The optimum pH for the Mucorales is well on the acid side of neutrality [5]. Nevertheless, some mucoraceous species have a pH optimum at quite high values. Another factor that may influence the quality of growth is light. These properties of wide ecological distribution, rapid growth and thermotolerance are of particular importance in causing human disease.

In many Mucorales that produce a sporangium, the sporangium wall breaks into a few separate, angular fragments under dry conditions [7]. Spores are readily blown away by wind. Usually, each cell fragment carries a load of spores. *Actinomucor elegans*, a frequent species of apparently world-wide distribution, behaves in a somewhat different manner. The whole spore mass enclosed by the sporangial wall slips off the smallish columella and is dispersed as a whole. Sporangiospores of the Mucorales group vary in dimensions. Specific differences in spore shape, size, and ornamentation were observed between *Rhizopus* taxa, and sometimes between strains. The spores of *Rhizopus* species typically are angular, sub-globose to ellipsoidal, with striations on the surface, and up to 8 µm in length. Robust spore ornamentation patterns can be linked to all different taxa of the *Rhizopus* group [8]. Ornamentation includes valleys and ridges running in parallel, granular plateaus, or smooth polar areas. The distribution of ornamentation patterns is related to spore shape, which either was regular, ranging from globose to ellipsoidal, or irregular. All of these features may influence dispersal in air and interaction with host mucosal surfaces.

Sporangiospores released by these fungi range from 3 to 11 μm in diameter, are easily aerosolised, and are readily dispersed throughout the outdoor and indoor environment [7]. Therefore, the sporangiospores of some species will be trapped in the upper airways whereas the smaller spores may emigrate to the alveoli. This is the major mode of transmission. The portals of entry of Mucorales are primarily the respiratory tract, the skin, and less frequently, the gut. These points of acquisition are very dependent on the ecological niche of the causative species. Also, spores can also be carried by insects, especially flies and spiders, as reviewed in [1,2].

## 3. Outdoor Habitats

### 3.1. Soil

A soil type is a taxonomic unit in soil science. All soils that share a certain set of well-defined properties form a distinctive soil type. Soil type is an abstract, technical term of soil classification—the science that deals with the systematic categorization of soils. Every soil of the world belongs to a specific soil type. Depending on local weather conditions such as temperature and wind currents, dust will be dispersed into the immediate surroundings of arable land, construction sites, and heavy excavation. Soil dust contributes a major fraction of the atmospheric particulate load in many regions of the globe. Wind-raised dusts from parched earth surfaces, for example, are probable causes of high levels of atmospheric soil dust.

There are currently approximately 3300 species of currently known soil fungi. Since soil is one of the most important biotopes for fungi, relatively high concentrations of fungal propagules are to be expected. Interestingly, very few studies have identified Mucormycetes at particular environmental sites where cases of mucormycosis have been reported from. For many years, there has been the view that most soil fungi are cosmopolitan and that species at a particular site are only selected by various soil parameters. The view has also been that most fungal species potentially spend part of their life in the soil. This has now been modified considerably, as many plant-parasitic species are never isolated from the soil [3]. This is particularly pertinent when considering the Mucorales.

Biological soil crusts are the community of organisms living at the surface of desert soils. Major components are cyanobacteria, green algae, microfungi, mosses, liverworts, and lichens. Mucorales do not appear to survive in this environment. Species of Mucorales have been cultured from geothermal soils in Yellowstone National Park, including species of *Lichtheimia* and *Cunninghamella*, especially in close proximity to *Dichanthelium langinosum*, a species of rosette grass native to North America. It is most common in the central and eastern United States, found in a variety of habitats, mostly in open, dry areas [2].

Disruption of the cutaneous barrier seems to be a prerequisite for the acquisition of cutaneous mucormycosis, with region-specific Mucorales being isolated from soil or sand at the geographical site of trauma as well as from the patient. Multiple studies mention soil contamination of soft tissue damage followed by mucormycosis, apparently in immunocompetent patients. Traumatic inoculation of spores can lead to extensive necrotic cutaneous infections. Cutaneous mucormycosis is an aggressive disease. It can lead to necrotizing fasciitis or to widespread disseminated infection. An illustrative case is where a patient who, while attempting to extinguish burning clothes, rolled in moist soil that was found to be contaminated with *Apophysomyces elegans* [2]. On the other hand, an example is a case of disseminated posttraumatic *Apophysomyces elegans* in an immunocompetent patient following soil inoculation [9]. *A. elegans* is traditionally found in warm climate soils, reviewed below.

What do we know about regional variation in the occurrence of Mucorales in soil? Prakash and colleagues sampled a total of 2188 soil samples from four provinces in Northern India and one in the South [10]. *Rhizopus oryzae* (24.6%) was the most commonly isolated followed by *Lichtheimia* spp. (23.2%), *Cunninghamella* spp. (21.7%), *Rhizopus microsporus* (14%) and *Apophysomyces* species complex (4.5%). The isolation of *Apophysomyces* spp. was specifically associated with low nitrogen content of the soil. The predominant species of *Apophysomyces* was *variabilis*. The importance of the isolation of *Apophysomyces* species from Indian soils is that this fungus is the most common agent causing mucormycosis in India and accounts for nearly 60% of the reported cases worldwide [11].

Mexico is the fifth largest producer of papaya worldwide and has recently reported problems with mucoraceous fungi affecting this crop. Cruz-Lachica and colleagues isolated mucormycetes from soil in regions of intensive papaya cultivation in Mexico [12]. Soil samples were collected from the states of Colima, Oaxaca and Veracruz. A total of 72 mucorales were isolated, including a number having medical importance, for example, *Rhizopus oryzae*, *Mucor circinelloides*, *Rhizopus microsporus*, and *Cunninghamella bertholletiae*.

The Atlantic Rainforest is a domain found all along the Brazilian coastline and is known for its biodiversity and endemic species of several fungal taxonomic groups. One order of fungi frequently found in the Atlantic Rainforest are the Mucorales. Little is known about the ecology of these fungi of the Atlantic Rainforest of Brazil. Lima and colleagues assessed the richness, diversity, frequency of occurrence and relative abundance of Mucorales in soil samples in an area of the Atlantic Rainforest located in the state of Pernambuco, Brazil [13]. A total of 32 soil samples were collected and eight genera and nine species of Mucorales were identified, and a new species of *Backusella* was reported. The authors note that the diversity of Mucorales was high when compared to results of other surveys of Mucorales in soil from the Atlantic Rainforest.

Does agricultural practice modify the structure of the soil fungal community? Silvestro and colleagues found that the composition of the fungal community varied according to different crops included in the cropping regimes [14]. Other factors were found to influence fungal diversity, such as season and sampling depth. Mixed cropping regimes including perennial pastures and one crop per year also promoted fungal diversity. Amongst numerous other moulds *Rhizopus stolonifera* was a predominant member of the soil mycobiota.

Mousavi and colleagues assessed the prevalence and diversity of species of Mucorales from soil samples collected in France [15]. A total of 170 soil samples were analysed. Forty-one isolates of Mucorales were recovered from 38 culture-positive samples. Among the isolates, 27 *Rhizopus arrhizus*, 11 *Mucor circinelloides*, one *Lichtheimia corymbifera*, one *Rhizopus microsporus* and one *Cunninghamella bertholletiae* were found. Positive soil samples came from cultivated fields but also from other types of soil such as flowerbeds. Mucorales were retrieved from samples obtained in different geographical regions of France. A novelty of this study is that voriconazole-containing medium improved the recovery of Mucorales compared with other media.

Are Mucorales found in desert soils and sands? Grishkan and colleagues examined the composition of the thermotolerant mycobiota in the soil of Israeli deserts and northern territories [16]. A total of 165 species from 82 genera were isolated including *Aspergillus fumigatus* and *A. niger* and *Rhizopus arrhyzus*.

Ziaee and colleagues collected 340 soil samples from public parks in Isfahan, Iran [17]. Four hundred pure colonies of *Mucor circinelloides*, *M. racemosus*, *M. plumbeus*, *Rhizomucor pusillus*, *Rhizopus arrhizus*, *R. stolonifera*, *Lichtheimia corymbifera*, *Cunninghamella bertholletiae* and *Mortierella wolfii* were identified by phenotypic and molecular identification methods. The genus *Rhizopus* (35.5%) was the most frequent isolate, followed by *Mucor* (32.5%) and *Rhizomucor* (27.5%).

A wide diversity of fungal taxa have been isolated from the sediments (bottom soils) of the White Sea—that is, species actively surviving in the littoral zone and at depths of 10–30 m. Khusnullina and colleagues found that even though the density of the bottom-soil fungi population was relatively low, the species diversity of species was rich including a small proportion of members of the Mucorales order indicating that these fungi can grow in sea water at low temperatures and varying oxygen levels [18]. This may have some relevance for cutaneous mucormycosis occurring after tsunamis and tidal waves.

A summary of soil types and global regions that support the survival of Mucorales is shown in Table 1.

### 3.2. Composting Vegetation

The elevated temperatures found in composting vegetation are selective for thermophilic species, such as some species of *Lichtheimia*, *Mucor*, *Rhizopus*, and *Rhizomucor*. In general, these fungi are unable to utilize cellulose and lignin. They are characterized by rapid germination. In general, composting is considered to be an aerobic process, suggesting high biological activity. It has been clearly demonstrated that the rise in temperature and the decomposition of composting plant materials is brought about by thermophilic microorganisms, including fungi. Temperature and changes in the available food supply probably exert the greatest influence in determining the species of organism comprising the population in a compost heap at any one time. Fungi, including *Aspergillus fumigatus* and Mucorales, play an important role in the decomposition of cellulose, lignin, and other more resistant materials, despite being confined primarily to the outer layers and becoming active only during the latter part of the composting period. Many opportunistic Mucorales are typical inhabitants of natural composts, tropical soils, and other heated materials. Indoor sites particularly associated with these fungi, therefore, may be those where humid organic material is exposed to heat, most notably within poorly maintained heating ducts and attached humidifier structures, in soils of potted plants (especially those placed in warm locations), and recycled vegetable matter in kitchens.

Hass and colleagues investigated the spectrum and the incidence of fungi in potting soils and compost [19]. The spectrum of fungi varied depending on the product. All soils showed a high proportion of *Aspergillus* and *Penicillium* species. The potting soils examined were dominated by *Penicillium spinulosum*, *P. simpliciccimum*, *Aspergillus fumigatus* and *Mortierella alpine*. *Mortierella* species are usually non-pathogenic for plants or animals and humans. A rare example of a pathogenic *Mortierella* species is *Mortierella wolfii*, which is until now the only pathogen of humans and other animals. *M. wolfii*, normally isolated from soil, rotten silage and similar substrates, causes bovine abortion, pneumonia and systemic mycosis. *Lichtheimia ramosa* was found in the potting compost of one plant. *Rhizopus* species were poorly represented in potting materials. The low burden of Mucormycotina species suggests that potting soils and compost do not constitute a particular risk to either immunocompromised or immunocompetent individuals.

### 3.3. Animal and Bird Excreta

Dung is a source of organic matter and a potential home for saprotrophs. From a fungal point of view, herbivore dung is the more interesting, since bacteria are largely responsible for the breakdown of carnivore and omnivore dung. Herbivore dung supports a wide variety of coprophilous fungi. The word coprophilous literally means “dung loving”. Herbivore dung typically contains plant material digested to varying extents.

The prevalence of Mucorales found growing on animal and bird excreta reflects the interest and activity of local research groups. De Souza and colleagues assessed and compared the Mucorales communities in dung from different species and breeds of herbivores in the semi-arid of Pernambuco, Brazil [20]. Samples of dung collected in the cities of Arcoverde, Serra Talhada and Sertânia were incubated in moist chambers in triplicate. Altogether, 24 taxa of Mucorales distributed in the genera *Absidia*, *Circinella*, *Cunninghamella*, *Lichtheimia*, *Mucor*, *Pilobolus*, *Rhizopus* and *Syncephalastrum* were identified. The highest species richness was found in sheep excrement. *Mucor circinelloides* f. *griseo-cyanus* was the most common taxon, followed by *M. ramosissimus*. The similarity of the composition of Mucorales species was greatest between the excrements of Guzerá and Sindi breeds (bovine).

Mucorales are found in pigeon excreta [21]. A study carried out in Karachi, Pakistan, analysed soil samples contaminated with pigeons’ excreta. The samples were collected from 20 different pigeon-feeding places in the city. These samples were processed for the isolation and identification of fungi by using standard conventional methods. One hundred and five samples were collected and analysed. A wide variety of fungal strains belonging to different genera of *Aspergillus*, *Rhizopus*, *Penicillium*, *Fusarium* and *Candida* were isolated. The authors concluded that the soil and air of places densely populated with pigeons are contaminated with fungal pathogens including agents of mucormycosis.

An agent of entomophthoromycosis, *Basidiobolus ranarum*, occurs with considerable regularity on the excrement of frogs, with the conidiophores projecting into the air. The conidiophore, which is positively phototrophic, arises from a single cell of the septate mycelium in the excrement. A single spore at the tip of an expanding conidiophore is propelled into the surrounding environment for a distance of 1–2 cm [7].

### 3.4. Specific Ecological Niches Revealed by Natural Disasters

Primary cutaneous mucormycosis is generally due to local trauma or inoculation (surgery, burns, motor vehicle-related trauma, the use of needles, knife wounds, insect or spider bites, and other types of trauma), while secondary infection is due to hematogenous dissemination of the organisms to the skin, as reviewed in [22]. Most patients with cutaneous mucormycosis have underlying diseases, such as haematological malignancies or diabetes mellitus, or solid organ transplantation, but a large proportion of them are immunocompetent. Cutaneous mucormycosis has also been reported to occur as a result of injury in a natural disaster, such as the tsunami that struck South-East Asia in 2004 [22].

Survivors of natural disasters with necrotising cutaneous mucormycosis have indirectly revealed specific ecological niches owing to wounds contaminated with water, soil, or debris [22]. An illustrative example is an outbreak of 13 cases of soft-tissue mucormycosis caused by *Apophysomyces trapeziformis* in individuals who were severely injured during a tornado in Joplin, Missouri in 2011 [23]. Whole-genome sequence typing (WGST) of isolates from case-patients’ wounds revealed four distinct but closely related strains of *A. trapeziformis*, suggesting that the cases were related to one or more environmental sources rather than a health care-associated source. The authors suggest that the environmental source(s) of *Apophysomyces* existed along the tornado path and that sporangiospores were dispersed from the mould growth on a variety of substrates and carried along with debris and inoculated with the debris after penetrating trauma. Genotyping of patient isolates suggested that there was either a single environmental source or a number of sources along the tornado path, for example, contaminated ponds. These findings prompted the authors to make a plea for further research on the ecology of *Apophysomyces* in order to gain a better understanding of its habitat and the environmental factors that affect its growth and diversity. This pathogen was found in a clinical case with a trauma injury and was also cultured from environmental samples [24]. *Apophysomyces* appears to grow in tropical and subtropical areas, and cases have been described from several states in the USA, different regions in India, Venezuela, Colombia, Mexico, Australia, Saudi Arabia, and Kuwait, suggesting that this fungus persists in soil or plant detritus in these areas. Further insight into ecological niches of *Apophysomyces* was revealed after the tsunami disaster in 2004, when victims were found to be infected with *Apophysomyces* contracted in Sri Lanka and Thailand [24]. *Saksenaea* has also been found causing necrotising cutaneous fasciitis following traumatic implantation by contaminated soil and water, and gross soft tissue trauma [11]. The genus comprises three species: *S. vasiformis* species complex, *S. erythrospora* and *S. oblongispora*.

## 4. Indoor Niches and Habitats Supporting the Growth of Mucorales

The mycobiota of indoor environments contains about 100–150 species [25,26]. Most species belong to the so-called anamorphic fungi, which have been known as Deuteromycetes, Hyphomycetes or Fungi Imperfecti. They can produce high concentrations of spores and compounds which can impact on health [25,26]). The most common and relevant fungal genera of the indoor mycobiota included *Aspergillus*, *Cladosporium*, *Fusarium*, *Penicillium*, *Scopulariopsis*, *Stachybotrys* [25,26]. The Mucorales are considered a minor member of the indoor mycobiota [25,26].

### 4.1. Indoor Air

Many fungi produce numerous spores or other propagules, and this explains how they can be present in high concentrations in the air [7]. Spores deposited in soil are a major source of contamination of the outdoor air. For the indoor air the outdoor air is an important source for fungal spores. The spore concentration, both outdoors and indoors, is dependent on the presence of vegetation and decaying substrates and climatic influences. However, there are very few data concerning the levels of Mucorales sporangiospores in outdoor and indoor air, especially in tropical and sub-tropical geographical areas where mucormycosis is particularly prevalent. The numbers of airborne Mucorales sporangiospores appear to depend on the presence of climatic conditions that favour growth and dispersal.

As has been noted in previous reviews, in spite of their abundance in soil, mucoraceous moulds do not figure prominently in the air spora [7]. An illustrative example is an extensive historical aeromycological study performed in Kansas, USA lasting over two years and sampling the air almost daily, as reviewed in [7]. Of 113,667 colonies obtained, only 194 belonged to Mucorales and of these 156 were *Rhizopus*. By way of contrast the figure for *Cladosporium* was 50,548. A survey of indoor and outdoor air in and around 17 homes in Cincinnati, Ohio, USA, using mould-specific quantitative PCR, failed to detect any pathogenic Mucorales [2]. During a two-year air-sampling survey in Barcelona, Spain, the following genera were found, in decreasing order: *Aureobasidum*, *Rhizopus*, *Mucor*, *Arthinium*, *Phoma*, *Fusarium*, *Trichoderma* and *Botrytis*, [2]. A one-year aeromycological study was conducted in the area of Zagreb, in order to establish seasonal variations in the composition and concentration of the aeromycota, [2]. Sampling was carried out at three locations at regular intervals using a Mas 100 Eco Air-sampler with Sabouraud-dextrose agar. The quantities of airborne fungi peaked during spring and summer (110–284 CFU/m^3^), and lower levels were detected in autumn and winter at each of the sampling sites (6–128 CFU/m^3^). In contrast to *Cladosporium*, *Pencillium* and *Alternaria*, very low levels of *Mucor* and *Rhizopus* were found.

In another large study of airborne indoor and fungal species and concentrations, Shelton and colleagues examined 9619 indoor samples from 1717 buildings located across the USA; these samples were collected during indoor air quality investigations performed from 1996 to 1998 [2]. For all buildings, air samples were collected with an Andersen N6 sampler. The culturable airborne fungal concentrations in indoor air were lower than those in outdoor air. The fungal levels were highest in the autumn and summer, and lowest in the winter and spring. Geographically, the highest fungal levels were found in the southwest, far west, and southeast regions. The most common culturable airborne fungi, both indoors and outdoors and in all seasons and regions, were *Cladosporium*, *Penicillium*, non-sporulating fungi, and *Aspergillus*. *Stachybotrys chartarum* was identified in the indoor air in 6% of the buildings studied. *Mucor* (species not specified), *Rhizopus* (species not specified) and *Cunninghamella* (species not specified) were detected in indoor air more than in outdoor air, although this varied from one geographical area of the USA to another. However, agents of mucomycosis were not included in the category of ‘common fungal types’.

### 4.2. Dust and Litter

A comparison of populations of mould species in homes in the UK and USA using mould-specific quantitative PCR did not detect any appreciable level of Mucorales in dust samples [2]. Notably absent from an extensive review of filamentous fungi isolated from settled dust in 369 homes in Ontario, Canada, are Mucorales [25]. This is a surprising finding, since a wide diversity of fungal species survive and propagate in settled dust if there is sufficient moisture.

Caetano and colleagues studied the prevalence of Mucorales in different workplaces (bakeries, swine farms, taxis, waste-sorting plants environments in Portugal) using an electrostatic dust collector (EDC) to sample air-conditioning filters, litter, and/or raw materials [27]. Air-conditioning filters from waste-sorting forklift trucks were the most heavily contaminated (35.7%). The predominant species were *Mucor* spp., *Rhizopus* spp. and *Rhizomucor* spp.

### 4.3. Manuscripts, Documents and Books

Large collections of cellulosic materials that are susceptible to microbial growth are found in libraries, archives and museum. These materials include paper, papyrus, cotton fabrics and photographic/cine film. Numerous investigations of the air spora in libraries have shown that Mucorales including *Rhizopus* species grow on paper, parchment and leather [25]. The extent of visible mould on these materials is not stated.

### 4.4. Building Materials

Mucorales do not appear to be common in buildings [25], suggesting that the various building materials used in house construction do not support the growth of these fungi, as compared with the wide profile of deuteromycetes found on damp substrates. A significant proportion of private residences, offices and workplaces are known to be damp, and estimates range from 20% to 50%. Numerous publications have reported the moisture levels required for the growth of fungi on construction, finishing or furnishing materials. *Rhizopus* species have a high moisture requirement for growth and are classified as hydrophilic. Members of the Mucorales are non-cellulolytic microorganisms, and do not have any enzymatic activity, even against the most susceptible forms of cellulose, and so are not very likely to be found on building materials. All of these observations suggest that house residents are not generally exposed to Mucorales in their home environment, apart from mould-contaminated food items such as bread and fruit (see below). This notion is supported by the absence of Mucorales in indoor air-sampling surveys. A review of the literature up to 2011 shows that members of the Mucorales group have not been isolated from cellulosic materials, gypseum board, urea-formaldehyde foam insulation, carpets, painted surfaces, ceramic tiles [25] apart from isolated reports reporting the isolation of Mucor from mould-affected building materials [25].

## 5. Food as an Ecological Niche for Mucorales

Mucorales are found contaminating a wide variety of food items [28]. Bouakline and colleagues reported the presence of Mucorales in pepper, regular tea, herbal tea, freeze-dried soups and sweet biscuits [28]. Common spoilage fungi in cultured dairy products, a variety of fresh produce, and baked goods. *Rhizopus* and *Mucor* spp. are part of the production of traditional fermented foods such as mould ripened cheeses and fermented soy products. Snyder and Worobo make a distinction between food contamination with Mucorales and those introduced and propagated during food processing [29]. Consumption of these food items is a potential risk for immunocompromised patients. It has been known for many years that *R. oryzae* readily contaminates grains, onions, various nuts, and stored seed potatoes [29].

The acquisition of mucormycosis through food among high-risk populations remains a concern in clinical settings. Gastrointestinal mucormycosis can arise from ingestion. However, there is the possibility that gastrointestinal tract infection originates from the lungs. Having said that fungal load on food products is a distinct possibility when evaluating consumer risk. *Mucor* and *Rhizopus* species have been isolated from bread and cheese. However, detection alone from the mycobiome of a food product represents a potentially lower degree of exposure than would consumption of a spoiled or mould fermented food where the fungal biomass has increased over several orders of magnitude.

Mucorales are found on food items. They are well recognised as spoilage agents of dairy products, cheese and yogurt. Samples isolated from Chobani yogurt that was voluntarily recalled in September 2013 have been found to contain *Mucor circinelloides*, which is associated with infections in immune-compromised people, as reviewed in [29]. Some Mucorales are unspecified parasites of the sappy tissues of higher plants. Species of *Rhizopus* commonly cause soft rot of fruit such as apples, plums and tomatoes. *Mucor* and *Rhizopus* also cause post-harvest spoilage of apples and pears, berries, tomatoes, aubergines, cherries, peaches, onions and potatoes, cabbages as well as various grains, salami and other low to medium water activity products.

An extensive study in France assessed the prevalence and diversity of human-pathogenic species of Mucorales in commercially available foodstuffs [30]. All food samples were purchased from January 2014 to May 2015. A total of 159 dried food samples including spices and herbs (*n* = 68), herbal tea (*n* = 19), cereals (*n* = 19), vegetables (*n* = 14), and other foodstuffs (*n* = 39) were analyzed. Each strain of Mucorales was identified phenotypically, and molecular identification was performed by ITS sequencing. From the 28 (17.6%) samples that were culture positive for Mucorales, 30 isolates were recovered. Among the isolates, 13 were identified as *Rhizopus arrhizus* var. *arrhizus*, 10 *R. arrhizus* var. *delemar*, two *Rhizopus microsporus*, one *Lichtheimia corymbifera*, three *Lichtheimia ramosa*, and one *Syncephalastrum racemosum*. Culture-positive samples originated from different countries (Europe, Asia) and brands. The samples most frequently contaminated by Mucorales were spices and herbs (19/68, 27.9%), followed by herbal tea (2/19, 10.5%), cereals (2/19, 10.5%), other food products (5/39, 12.8%). This extensive study showed that human-pathogenic Mucorales were frequently recovered from commercially available foodstuffs in France with a large diversity of species. The authors highlight the potential danger of exposure to Mucorales present in food by immunocompromised patients.

## 6. Water

### 6.1. Drinking Water

The diversity of fungi in drinking water depends on organic matter from natural and anthropogenic sources, as reviewed in [31]. The concentration of organic matter in water depends on the location and the surface area of bodies of water. Surface water contains a high biomass and a rich diversity of plant degrading filamentous fungi [31]. In Europe, the majority of fungal species that have been isolated from domestic water belong to the Ascomycetes but also from the Mucorales including *Absidia* spp., *Lichtheimia corymbifera*, *Mortierella* species, *Mucor* spp., *Mucor circinelloides*, *Rhizomucor* spp., *Rhizopus oryzae*, and *Synchepalastrum racemosum* [31].

### 6.2. Marine Environments

Are Mucorales found in marine environments? An important question to pose since cases of cutaneous mucormycosis caused by Mucorales following traumatic injury during tsunamis has been reported [22]. In an exhaustive review of fungi isolated from marine habitats Mucorales are not represented [32]. Mucorales are primarily terrestrial fungi are rarely found growing in truly marine habitats. i.e., where the ambient water is in the order of 30% saline. Ribes and colleagues do not record any reports of Mucorales being isolated from marine environments [1]. However, the finding of specific Mucorales in the sediments of the White Sea suggests otherwise [18].

## 7. Hospital Environments

The isolation of Mucorales from hospital environments has been reviewed previously [1,2]. A number of studies have followed since. Qudiesat and collaborators evaluated the quality of air as well as the level of airborne moulds in two hospitals [33]. They found that in both hospitals *Aspergillus* spp., *Penicillium* spp., *Rhizopus* spp. and *Alternaria* spp. were the predominant fungal species found in hospital air. Interestingly, the authors also verified that the concentration of fungi and bacteria in the air of both environments were influenced by human occupation [33].

An extensive treatise on health care-associated mucormycosis has been presented by Rammaert and colleagues [34]. These authors make it clear that for many of the cases and outbreaks of mucormycosis that have been reviewed, it is not always clear where the infecting organism was acquired from. However, a number of culprit sources of contamination are clear, including adhesives containing karaya paste used for ostomy bags, wooden tongue depressors used for the mixing of oral treatments, contaminated catheters, various dressings, and building construction. The review authors do suggest that it is often difficult to definitively implicate a source of infection. Only in a few studies was systematic sampling of the implicated device carried out. However, hospital-acquired infections and outbreaks do suggest that the Mucorales can grow on a variety of substrates.

Mucormycosis outbreaks among hospitalized patients in the United States have become more frequent in recent years. Three recent outbreaks in hospitals reveal specific environments that are found in hospitals and lead to such outbreaks, as reviewed in [22]. A cluster of mucormycosis occurred among four solid organ transplant recipients in Pennsylvania in May 2014–September 2015. Although the patients lacked a clinical indication for negative-pressure isolation, three of the four received care in the same negative-pressure isolation room suggesting that contaminated air was drawn into the room. This investigation led to the important finding that hospitals should avoid placing such high-risk patients in negative-pressure rooms unless indicated. In another recent cluster, in March–June 2015, five mucormycosis cases were observed among bone marrow transplant recipients at a hospital in Colorado where construction was ongoing in an adjacent unit. A definitive source could not be identified, but mould that accumulated in the wall due to a slow water leak may have played a role. Water damage and mould growth is a recurrent theme in hospital outbreaks.

In July–November 2014, five patients with haematological malignancies developed rhinocerebral mucormycosis at a hospital in Kansas, as reviewed in [22]. The infections, caused by several different mucormycetes, coincided with nearby construction, and the case-patients were placed in rooms that shared a hallway with construction traffic. Hospital construction is a well-recognized risk factor for mould infection, and infection control risk assessments are recommended before beginning construction or other activities that may generate dust or moisture, potentially dispersing mould spores, as reviewed in [22]. Although much care was taken to implement control measures within the construction zone in this outbreak, similar precautions were not taken outside the construction zone, with workers using the same hallway as patients on a haematology/oncology ward. This outbreak highlighted the importance of ensuring complete separation of patients and construction activities and personnel even outside of construction areas.

These recent mucormycosis outbreaks highlight the need to pay close attention to the hospital environment. Investigating these outbreaks presents several challenges. First, since there is no routine surveillance for mould infections, and cases in an outbreak often occur over several months, it is difficult to determine whether and when and an outbreak is occurring. Second, numbers of outbreak-associated cases are often small, and case-control studies to determine common exposure may not yield any significant results. Lastly, more research is needed to understand the optimal indoor environments in hospitals and how to achieve and maintain those standards. In the United States, there are currently no standards for airborne mould concentrations, and the value of routine hospital air sampling is unclear, though a recent study from Spain identified high levels of airborne *Aspergillus fumigatus* in association with aspergillosis cases in heart surgery patients, as reviewed in [22].

Recently, a number of outbreaks were linked to contaminated hospital linens or laundry carts and bins [35,36,37,38]. The full extent to which contaminated linens contribute to the burden of hospital-acquired mucormycosis is unknown and may be under-appreciated. Moreover, it is unknown whether contaminated linens account for sporadic cases of mucormycosis. Sundermann and colleagues performed fungal cultures on freshly laundered linens at 15 transplant and cancer hospitals [38]. At 33% of hospitals the lines were visibly unclean. At 20% of hospitals Mucorales were recovered from >10% of linens. In the United States at least, clinical and infection control teams should be aware that linen supplied to their hospitals may be contaminated with Mucorales, highlighting that a minimum requirement is that periodic visual inspections of linens and linen bins for general cleanliness should be performed. It is recognized that a number of factors may promote the contamination of hospital linen including inadequate drying, temperature, poor off-site laundry facilities, season, and the occurrence of Mucorales in the outdoor environment.

## 8. Conclusions

Mucorales here, Mucorales there, Mucorales everywhere? Yes and no. Even though the Mucorales appear to be ubiquitous in a number of environments, as outlined here, it is unclear how patients with mucormycosis acquire their infection. Questions still remain regarding the frequency of exposure, the influence of geographical location, and what the minimum effective dose is. Consequently, there are few means of preventing mucormycosis. Better understanding of the source of infection is key for preventing infection in vulnerable patients.

## Figures and Tables

**Table 1 jof-06-00004-t001:** Diversity of Mucorales in various soil types representing different soil orders.

Geographical Location	Type of Soil	Taxa/Species Isolated
India, Haryana, Punjab, Himachal Pradesh, Tamil Nadu [10]	Agricultural, non-agricultural, low nitrogen content	*Rhizopus arrhizus*
		*Lichtheimia* spp.
		*Cunninghamella* species
		*Rhizopus microsporus*
		*Apophysomyces* species
		*Apophysomyces viariabilis*
Mexico [12]	Intensive papaya producing orchards in Colima, Oaxaca and Veracruz states	*Gilbertella persicaria*
		*Rhizopus oryzae*
		*Rhizopus stolonifera*
		*Mucor circinelloides*
		*Mucor hiemalis*
		*Choanephora cucurbitarum*
		*Mucor ellipsoideus*
		*Rhizopus homothallicus*
		*Rhizopus microspores*
		*Rhizopus schipperae*
		*Lichtheimia*
		*Gongronella butleri*
		*Cunninghamella bertholletia*
		*Cunninghamella blakesleena*
Brazil [13]	Semi-arid	*Absidia (Lichtheimia)*
		*Cunninghamella*
		*Gongronella*
		*Mucor*
		*Rhizopus*
		*Synchephalastrum*
Argentina, Buenos Aires [14]	No tillage for 13 years, Petrocalcic Argiudoll type and topsoil with a sandy clay loam texture, different cropping regimens	*Rhizopus stolonifer*
France, different regions [15]	Arable fields, flower beds	*Rhizopus arrhizus*
		*Mucor circinelloides*
		*Lichtheimia corymbifera*
		*Rhizopus microsporus*
		*Cunninghamella bertholletiae*
Israel, Negev Desert [16]	Desert sand, loess soils, stony debris	*Mortierella*
		*Rhizopus* spp.
Iran, Isfahan [17]	Parks, gardens	*Mucor circinelloides*
		*M. racemosus*
		*M. plumbeus*
		*Rhizopmucor pusillus*
		*Rhizopus arrhizus*
		*R. stolonifera*
		*Lichtheimia corymbifera*
		*Cunninghamella bertholletiae*
		*Mortierella wolfii*
Israel, White Sea [18]	Bottom soils, sediments	*Mucor hiemalis*

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
