# Peer review of "Biotic Environments Supporting the Persistence of Clinically Relevant Mucormycetes"

_jof, 2019, doi:10.3390/jof6010004_

Round 1
Reviewer 1 Report
The presented manuscript is a large review of recent researches about Mucorales ecology. The Authors focus on ecological studies published after 2000 which included results about distribution or presence of Mucorales representatives in particular localities/niches. Although the study is interesting and it brings up to date the knowledge about Mucoralean ecology, it lacks clear conclusions or summary. Therefore, I recommend the revision of article structure, focusing more on drawing joint conclusions than on enumeration of subsequent papers.
In introduction part I lack at least short information about current systematics of Mucoromycetes. Without that it is not clear what fungi are supposed to be included in the study. The second problem is structure of chapters, for example I can’t see the reason why workplaces are taken out of “indoor niches”. I also have troubles to see why composting vegetation and compost are treated separately. In my opinion putting some things together could help in making some more general conclusions.
Some chapters have quite narrow title, like “how are Mucorales transmitted?”, while there is not a lot of information in it. I would prefer to reorganize the first “Introduction” chapter to something like “general information on Mucoromycetes”, including information on systematics, transmission, growth conditions etc. Then I would see the review of subsequent niches like: water, air (indoor and outdoor), soil, plant substrates, animal related substrates. In my opinion, the deeply thought-out hierarchical structure of the chapters is crucial for creating a general picture of the ecology of presented group. It would be also better to focus only on this group while describing some papers. For example lines 261-268 are focused on Mucorales on pigeon excreta. There is a lot of information about number of analysed samples but this information is not crucial for understanding of Mucoralean ecology. Similarly information about all other taxa detected in each particular study is not very relevant for Mucorales ecological niches description.
There are many editorial mistakes in the text: various types of fonts (line 137), incomplete citations (lines 212 and 396), lack of italics in species names (eg. lines 333, 362, 365), lines switches (lines 331 and 337). The Authors should also try to use rather scientific than popular science language.
Author Response
1.
Comments and Suggestions for Authors
The presented manuscript is a large review of recent researches about Mucorales ecology. The Authors focus on ecological studies published after 2000 which included results about distribution or presence of Mucorales representatives in particular localities/niches. Although the study is interesting and it brings up to date the knowledge about Mucoralean ecology, it lacks clear conclusions or summary. Therefore, I recommend the revision of article structure, focusing more on drawing joint conclusions than on enumeration of subsequent papers.
We thank you for comments and suggestions. We have clarified the scope. The new title describes the intended content better. We have not attempted to provide an exhaustive review of all the ecological niches but have selected illustrative examples for each environments which have not been covered in any great depth by previous reviewers. Where available, we have referred to previous reviews, and not the original source material to avoid cluttering the text with numerous citations.
We have not attempted to provide an exhaustive review of all the ecological niches but have selected examples of environments which have not been covered in any great depth by previous reviewers.
In introduction part I lack at least short information about current systematics of Mucoromycetes. Without that it is not clear what fungi are supposed to be included in the study.
Our review is not published in isolation but a part of a special collection of papers/reviews in the Journal of Fungi hence we have simply referred to other papers on taxonomy/systematics in the introductory sections of our review
The second problem is structure of chapters, for example I can’t see the reason why workplaces are taken out of “indoor niches”. I also have troubles to see why composting vegetation and compost are treated separately. In my opinion putting some things together could help in making some more general conclusions.
We have restructured the text to accommodate the reviewers concerns
Some chapters have quite narrow title, like “how are Mucorales transmitted?”, while there is not a lot of information in it. I would prefer to reorganize the first “Introduction” chapter to something like “general information on Mucoromycetes”, including information on systematics, transmission, growth conditions etc. Then I would see the review of subsequent niches like: water, air (indoor and outdoor), soil, plant substrates, animal related substrates. In my opinion, the deeply thought-out hierarchical structure of the chapters is crucial for creating a general picture of the ecology of presented group. It would be also better to focus only on this group while describing some papers. For example lines 261-268 are focused on Mucorales on pigeon excreta. There is a lot of information about number of analysed samples but this information is not crucial for understanding of Mucoralean ecology. Similarly information about all other taxa detected in each particular study is not very relevant for Mucorales ecological niches description.
We have taken these comments into account. Our aim was not to focus on the ecology of the Mucorales as such but to suggest exposure pathways, whether direct contact or by inhalation. Furthermore, we have selected illustrative examples of where Mucorales are known to survive, and what species were found. We have not attempted to review every paper within each niche. The focus is on recently published reports highlighting specific environments.
There are many editorial mistakes in the text: various types of fonts (line 137), incomplete citations (lines 212 and 396), lack of italics in species names (eg. lines 333, 362, 365), lines switches (lines 331 and 337). The Authors should also try to use rather scientific than popular science language.
Corrected and modified. Point about journalistic language has been addressed in most cases.
Reviewer 2 Report
Richardson & Rautemaa-Richardson present a review of the ecological niches of mucormycetes (Mucorales), aiming to update this previously reviewed topic and to examine specialized exposure scenarios such as natural disasters. This is a timely subject, since recent comprehensive reviews of the ecology of Mucorales are lacking and a number of exciting studies have been published in the past decade, e.g. with regards to natural-disaster-related Mucorales infections, combat-related myocutaneous mucormycosis, or the identification of antifungal-resistant Mucorales in indoor environments.
Unfortunately, this review article is not very well composed, and the scope envisioned by the authors remains unclear. The abstract and statement in line 32-33 suggests a focus on recent literature to build upon earlier reviews (refs 1 and 2), whereas in line 35 the authors claim to “examine the literature old and new”. For some sections such as chapters 2, 3 or 5.1, the authors provide lengthy summaries of older literature (without references to original research), whereas other chapters (e.g. 5.3 or 9) cite exclusively new references (often few or even just a single one). This should be uniformly dealt with throughout the article.
I would clearly recommend to consider this article as an update on the ecology of Mucorales (this may be highlighted in the title). The authors may begin each chapter with a brief summary (1-2 short paragraphs) of the knowledge presented in previous reviews, followed by the questions addressed by the literature in the past decade. Even with a focus on literature from the past 10-20 years, the article lacks dozens of relevant references, and is therefore not able to provide a comprehensive report of the recent advances in this field. Throughout the article, the authors often refer to other review articles instead of citing relevant original research.
Therefore, a profound revision of this article would be warranted to improve its focus and clarity. A number of additional relevant studies needs to be incorporated, and the authors should ensure that the review is well-tied to original individual references instead of previous review articles.
Specific Points:
· The abstract mentions “theatres of war” as a specialized niche, but no further discussion of literature pertaining to combat-related injuries has been provided in the review. Indeed, this would represent a highly interesting addition to this article. There has been plenty of recent literature on this topic including several original research articles from the Trauma Infectious Disease Outcomes Study Group (TIDOS).
· Chapter 4.2 should be substantially expanded and original articles should be cited (e.g. Neblett Fanfair et al., 2012; Etienne et al., 2011; Andresen et al., 2005).
· There are several paragraphs or even entire chapters without a single reference (e.g. 4.3) or review citations only (e.g. chapter 10 and most of chapter 3).
· Chapter 9 should either be removed from the article or significantly expanded. There has been further literature in the past decade on Mucorales exposure in workplaces, e.g. Lawniczek-Wałczyk et al., 2012, Faerden et al., 2014, Rohr et al., 2015, and also additional cases of (possibly) work-related mucormycosis, e.g. Rabie & Althaqafi, 2012. In addition, while the article clearly focusses on sources of invasive infection, chapter 9 should include a brief statement that hypersensitivity syndromes (such as hypersensitivity pneumonitis or asthma) are far more commonly associated with occupational exposure to Mucorales than invasive mucormycosis.
· The concluding chapter 11 is not very convincing. It should more aptly round off the article, focusing on the novel aspects we have learned about the Mucorales ecology in the past 10-20 years (e.g. their relevance in trauma-related infections of otherwise non-immunocompromised individuals).
· Chapters 2 and 4.1 contain many technical terms that are not necessarily required for the understanding of Mucorales ecology. This became particularly obvious for the lengthy definition of the term “soil type” in lines 108-111 without subsequently using the term in the article.
· Lines 80-97 should be summarized more concisely and included in chapter 3.
· Lines 269-273 should be deleted.
· Some of the (rhetorical) questions come across as a distraction, especially when placed in the middle of the chapters or when using interleaved questions while still answering the one raised before (especially in chapter 4). This should be mitigated to make the article more easily accessible and more pleasant to read.
Author Response
2.
Richardson & Rautemaa-Richardson present a review of the ecological niches of mucormycetes (Mucorales), aiming to update this previously reviewed topic and to examine specialized exposure scenarios such as natural disasters. This is a timely subject, since recent comprehensive reviews of the ecology of Mucorales are lacking and a number of exciting studies have been published in the past decade, e.g. with regards to natural-disaster-related Mucorales infections, combat-related myocutaneous mucormycosis, or the identification of antifungal-resistant Mucorales in indoor environments.
Unfortunately, this review article is not very well composed, and the scope envisioned by the authors remains unclear. The abstract and statement in line 32-33 suggests a focus on recent literature to build upon earlier reviews (refs 1 and 2), whereas in line 35 the authors claim to “examine the literature old and new”. For some sections such as chapters 2, 3 or 5.1, the authors provide lengthy summaries of older literature (without references to original research), whereas other chapters (e.g. 5.3 or 9) cite exclusively new references (often few or even just a single one). This should be uniformly dealt with throughout the article.
Comments taken into account and accommodated. We have clarified the scope. The new title describes the intended and restricted content better. We have not attempted to provide an exhaustive review of all the ecological niches but have selected illustrative examples for each environment which have not been covered in any great depth by previous reviewers. In some instances, we have referred to previous reviews, and not the original source material to avoid cluttering the text with numerous citations.
I would clearly recommend to consider this article as an update on the ecology of Mucorales (this may be highlighted in the title). The authors may begin each chapter with a brief summary (1-2 short paragraphs) of the knowledge presented in previous reviews, followed by the questions addressed by the literature in the past decade. Even with a focus on literature from the past 10-20 years, the article lacks dozens of relevant references, and is therefore not able to provide a comprehensive report of the recent advances in this field. Throughout the article, the authors often refer to other review articles instead of citing relevant original research.
To a large extent we have accommodated these concerns. We have not attempted to provide a comprehensive report of the recent advances of the field but to give selected illustrative examples for the environments that support the growth of the Mucorales. The revised title reflects the focus of our mini-review.
Therefore, a profound revision of this article would be warranted to improve its focus and clarity. A number of additional relevant studies needs to be incorporated, and the authors should ensure that the review is well-tied to original individual references instead of previous review articles.
We trust that our brief text has been revised satisfactory. Please see above our response to the point about citing original articles. Our previous review (reference 2) puts these older reports into context, hence the reason why we have referred to the 2009 review.
Specific Points:
The abstract mentions “theatres of war” as a specialized niche, but no further discussion of literature pertaining to combat-related injuries has been provided in the review. Indeed, this would represent a highly interesting addition to this article. There has been plenty of recent literature on this topic including several original research articles from the Trauma Infectious Disease Outcomes Study Group (TIDOS).Our question: do these papers suggest or explore the environmental sources where these infections occurred? We are aware of the papers suggested by the reviewer but the authors do not explore, but only surmise, the presence of the infecting species in the areas where mucormycosis was contracted/acquired.
Chapter 4.2 should be substantially expanded and original articles should be cited (e.g. Neblett Fanfair et al., 2012; Etienne et al., 2011; Andresen et al., 2005).We have reviewed the Neblett Fanfair report as a good example of the ensuing infections with Apophysomyces being a surrogate/proxy for the very likely contamination of the tornado area with this fungus.
There are several paragraphs or even entire chapters without a single reference (e.g. 4.3) or review citations only (e.g. chapter 10 and most of chapter 3).Citations added
Chapter 9 should either be removed from the article or significantly expanded. There has been further literature in the past decade on Mucorales exposure in workplaces, e.g. Lawniczek-Wałczyk et al., 2012, Faerden et al., 2014, Rohr et al., 2015, and also additional cases of (possibly) work-related mucormycosis, e.g. Rabie & Althaqafi, 2012.We thank the reviewer for bringing these studies to our attention. We have deleted this section. Our review now focuses on the growth of Mucorales on specific substrates.
In addition, while the article clearly focusses on sources of invasive infection, chapter 9 should include a brief statement that hypersensitivity syndromes (such as hypersensitivity pneumonitis or asthma) are far more commonly associated with occupational exposure to Mucorales than invasive mucormycosis.
We thank the reviewer for bringing this point to our attention.. Our review now focuses on the growth of Mucorales on specific substrates.
The concluding chapter 11 is not very convincing. It should more aptly round off the article, focusing on the novel aspects we have learned about the Mucorales ecology in the past 10-20 years (e.g. their relevance in trauma-related infections of otherwise non-immunocompromised individuals).
Please see our responses above where we have clarified the intention of our mini-review
Chapters 2 and 4.1 contain many technical terms that are not necessarily required for the understanding of Mucorales ecology. This became particularly obvious for the lengthy definition of the term “soil type” in lines 108-111 without subsequently using the term in the article.We describe soil types in relation to the growth and persistence of Mucorales in an attempt to surmise where these fungi might and might not grow.
Lines 80-97 should be summarized more concisely and included in chapter 3.Text modiifed
Lines 269-273 should be deleted.Done
Some of the (rhetorical) questions come across as a distraction, especially when placed in the middle of the chapters or when using interleaved questions while still answering the one raised before (especially in chapter 4). This should be mitigated to make the article more easily accessible and more pleasant to read.Deleted
Reviewer 3 Report
There is indeed a need to review the ecology of mucoralean fungi, also with respoect to those features that enable some of them to colonise animals and humans. There are, however, so many points of criticism that I cannot recommend this manuscript for publication. Some details that hopefully help to prepare a better manuscript in the future are given below.
* The most general critical point: The interpretation of "Ecology" does not seem adequate. Ecology, in the context of Mucor-related fungi or other groups, can never be focused as much on medical aspects as the authors do in this review. Indeed, Mucor-species are are only sparsely found as growing mycelia on or in man or other mammalian hosts. If they are found, they have a strong tendency towards untypical morphological structures. They are predominantly found on plant litter or in soil and indeed on sugar-containing fruit or, many of them, on dung of herbivorous animals. For such biotopes, there is plenty of literature on their ecology and the neccessary biochemical and physiological prerequisites, indeed muchg more than the authors cover in their contribution.
* There are also several, although not too many, Mucor-like fungi that grow predominantly on proteins by expressing a variety of interesting proteases (some of them being used for biotechnical purposes). It would be highly interesting - with respect to the ecology aspect, addressed in the title - to review those publications devoted to the biochemical and physiological potential of Mucors with relationships to potential aggressivity towards animals including man. This manuscript does not appropriately cover the literature in this and indeed other important aspects.
* If, as in line 22, the authors start their review with the question "What do we know so far?", they should answer it, which is indeed the major intention of a review. It is by no means sufficient, to cite a single publication (Ribes et al.) that is claimed to cover "exhaustively" "the natural habitats of mucoraceous moulds". This is certainly not true. The review cited is exclusively devoted to zygomycetes in human disease, which is already made clear by the title. From a new review on "Ecological niches of zygomycetes", a much broader coverage of the topic must be expected. There is nothing wrong with discussing ecology of zygomycetes in the context of human disease, but as humans are definitely not the major natural biotopes, where zygomycetes develop and propagate, these medical aspects have, very carefully, to be brought together with the ecology of these fungi.
* Indeed, the manuscript offers a good basis for a general discussion of niches that are colonised by Mucor-related fungi. But the prerequisits that favor Mucor-colonisation must be described and analysed in more depth than the authors do. There is much more literature that needs to be included approriately.
* In several important parts of the manuscript, the authors are too general and do not properly substantiate those claims. Line 99 ff: "Glucose is the main carbon source for growth. Like other fungi, Mucorales have a requirement for proteins and a range of organic and inorganic substances. The pH of a particular environment can markedly affect the rate of hyphal extension, growth, and sporulation. Another factor that may influence the quality of growth is light." The very important points for growth and propagation of zygomycetes must certainly be carefully substantiated. Light has been studied in depth in this group of fungi, and there are many effects on the biology of zygomycetes. Glucose availability is an interesting question that needs to be analysed in depth. Simple sugars are indeed preferred by many zygomycetes, and their availabilty might explain a lot about the niches colonised. The requirement for proteins is a very interesting and important point, too. Indeed, some zygomycetes have auxotrophies for certain amino acids, and thus, rely on peptides or proteins, and certainly on proteases. Others are completely prototrophic. With respect to ecology, these points, and others, must be addressed in detail. What do the authos mean exactly with "organic and inorganic substrates"? They should substantiate this. How do zygomycetes and ascomycetes (the major field of expertise of the authors) differ in these respects?
* The descriptions of hyphal development, differentiation of sporangiospores etc. need instructive drawings. With respect to airborne dissemination of spores, the authors need to provide a more comprehensive survey of spore features and must certainly differentiate These features from those offered by Aspergilli and similar moulds.
* Sometimes, especially for Rhizopus, the authors give details of spore sizes, ornamentation etc. If These features are important for colonisation of ecological niches, they must give these details also for the other genera (and maybe species) mentioned. If they do and compare these data, they could do a much better job than simply to state that "These features may influence dispersal in air and interaction with host mucosal surfaces" (line 97).
* Overall, the authors have collected a basis for reviewing and interpreting the ecology of mucoralean fungi, but taken together, with the points of criticism already mentioned and the low coverage of important literature, the manuscript cannot be recommended for publication - at least not at the present stage of collecting and interpreting the available literature.
Author Response
3.
Comments and Suggestions for Authors
There is indeed a need to review the ecology of mucoralean fungi, also with respoect to those features that enable some of them to colonise animals and humans. There are, however, so many points of criticism that I cannot recommend this manuscript for publication. Some details that hopefully help to prepare a better manuscript in the future are given below.
* The most general critical point: The interpretation of "Ecology" does not seem adequate. Ecology, in the context of Mucor-related fungi or other groups, can never be focused as much on medical aspects as the authors do in this review. Indeed, Mucor-species are are only sparsely found as growing mycelia on or in man or other mammalian hosts. If they are found, they have a strong tendency towards untypical morphological structures. They are predominantly found on plant litter or in soil and indeed on sugar-containing fruit or, many of them, on dung of herbivorous animals. For such biotopes, there is plenty of literature on their ecology and the neccessary biochemical and physiological prerequisites, indeed muchg more than the authors cover in their contribution.
We have modified the terminology to better reflect the aim of this mini-review
* There are also several, although not too many, Mucor-like fungi that grow predominantly on proteins by expressing a variety of interesting proteases (some of them being used for biotechnical purposes). It would be highly interesting - with respect to the ecology aspect, addressed in the title - to review those publications devoted to the biochemical and physiological potential of Mucors with relationships to potential aggressivity towards animals including man. This manuscript does not appropriately cover the literature in this and indeed other important aspects.
We have clarified the scope. The new title describes the intended content better. We have not attempted to provide an exhaustive review of all the ecological niches but have selected illustrative examples for each environments which have not been covered in any great depth by previous reviewers. Where available, we have referred to previous reviews, and not the original source material to avoid cluttering the text with numerous citations.
* If, as in line 22, the authors start their review with the question "What do we know so far?", they should answer it, which is indeed the major intention of a review. It is by no means sufficient, to cite a single publication (Ribes et al.) that is claimed to cover "exhaustively" "the natural habitats of mucoraceous moulds". This is certainly not true. The review cited is exclusively devoted to zygomycetes in human disease, which is already made clear by the title. From a new review on "Ecological niches of zygomycetes", a much broader coverage of the topic must be expected. There is nothing wrong with discussing ecology of zygomycetes in the context of human disease, but as humans are definitely not the major natural biotopes, where zygomycetes develop and propagate, these medical aspects have, very carefully, to be brought together with the ecology of these fungi.
We have addressed this buy revising the title and the content of the review
* Indeed, the manuscript offers a good basis for a general discussion of niches that are colonised by Mucor-related fungi. But the prerequisits that favor Mucor-colonisation must be described and analysed in more depth than the authors do. There is much more literature that needs to be included approriately.
We completely agree with this. But as said, we have not attempted to provide an exhaustive review of all the ecological niches but have selected illustrative examples for each environments which have not been covered in any great depth by previous reviewers. We have also provided, and referenced, key points pertaining to the characteristics of substrates that support the growth of Mucorales.
* In several important parts of the manuscript, the authors are too general and do not properly substantiate those claims. Line 99 ff: "Glucose is the main carbon source for growth. Like other fungi, Mucorales have a requirement for proteins and a range of organic and inorganic substances. The pH of a particular environment can markedly affect the rate of hyphal extension, growth, and sporulation. Another factor that may influence the quality of growth is light." The very important points for growth and propagation of zygomycetes must certainly be carefully substantiated. Light has been studied in depth in this group of fungi, and there are many effects on the biology of zygomycetes. Glucose availability is an interesting question that needs to be analysed in depth. Simple sugars are indeed preferred by many zygomycetes, and their availabilty might explain a lot about the niches colonised. The requirement for proteins is a very interesting and important point, too. Indeed, some zygomycetes have auxotrophies for certain amino acids, and thus, rely on peptides or proteins, and certainly on proteases. Others are completely prototrophic. With respect to ecology, these points, and others, must be addressed in detail. What do the authos mean exactly with "organic and inorganic substrates"? They should substantiate this. How do zygomycetes and ascomycetes (the major field of expertise of the authors) differ in these respects?
We have addressed this by referencing a treatise by Ingold on these points.
* The descriptions of hyphal development, differentiation of sporangiospores etc. need instructive drawings. With respect to airborne dissemination of spores, the authors need to provide a more comprehensive survey of spore features and must certainly differentiate These features from those offered by Aspergilli and similar moulds.
The emphasis of our review has markedly changed and we have significantly removed any detailed discussion on ecological aspects pertaining to the growth of the Mucorales.
* Sometimes, especially for Rhizopus, the authors give details of spore sizes, ornamentation etc. If These features are important for colonisation of ecological niches, they must give these details also for the other genera (and maybe species) mentioned. If they do and compare these data, they could do a much better job than simply to state that "These features may influence dispersal in air and interaction with host mucosal surfaces" (line 97).
This has been addressed – we believe that the size and ornamentation of Mucorales sporangiospoes are important for impaction on mucosal surfaces or adherence to exposed soft tissue.
* Overall, the authors have collected a basis for reviewing and interpreting the ecology of mucoralean fungi, but taken together, with the points of criticism already mentioned and the low coverage of important literature, the manuscript cannot be recommended for publication - at least not at the present stage of collecting and interpreting the available literature.
Please see our responses above. We have addressed the points raised in line with the clarified scope of this review.
Reviewer 4 Report
The authors have undertaken an ambitious work by approaching numerous environmental aspects of mucorales and mucormycosis. They have identified interesting studies on ecological niches of mucorales, but the review would be improved by performing a more systematic collection of data in the different studies that were picked. The contribution of the present manuscript needs to be clarified in the paragraphs which rely on previous reviews.
General comments:
References should be added in the paragraphs which contain none (e.g. no reference is cited in lines 32-51, lines 218-232, lines 353-359, etc…).
The authors have chosen to do only text description of studies. However, it would be very helpful for the reader if they could add tables to describe the reported studies. For example, for studies on soil, such a table could include geographic area, climate/soil type/cropping regimes, number of soil samples collected, mucorales species isolated, proportion when compared to other filamentous fungi, individual frequency,… It would allow a more systematic collection of data, and such tables could be useful to make further comparisons of frequencies and to identify factors associated with some mucorales genera or species.
For some paragraphs, the text nearly only cites previous reviews. The authors could alternatively present these reviews at the beginning of the manuscript and give their arguments for not including the aspects which have already been reviewed. Otherwise, the concerned paragraphs need to be updated (see below), and references of the original studies should be cited.
Specific comments:
Lines 22 and 32: Please correct “What do know so far?”, and “The present review focusses on THE what we have learnt”
Lines 66-68: the sentence seems contradictory. Please clarify if Lichtheimia species are xerophilic or hydrophilic.
Line 147: Please clarify the sentence, for example by adding “Apophysomyces cases in India » before “accounts for nearly 60% of the reported cases worldwide”. Otherwise it seems that this fungus “accounts for nearly 60% of the reported mucormycosis cases worldwide”.
Lines 203-208 and lines 215-216: Please cite the original references of these studies. Further details on the 2004 tsunami would be needed here.
Lines 209-215: This part seems not to be adequate in this paragraph, where rather context, number of cases, and/or species isolated in natural disasters are expected.
Lines 275-312: Although this paragraph is entitled “Indoor air”, is seems that most data concern outdoor air. Moreover this paragraph mainly relies on a review that was published in 2009. Is there any recent data that could be added? The authors give a lot of information that does not concern mucorales, but, for example in lines 310-311, it would be interesting to have the frequencies of mucorales species. Please cite the original references of studies.
Line 402: Healthcare-associated mucormycosis have also been reviewed by Rammaert et al. in 2012. Building construction is mainly underlined by the authors in this paragraph, but other nosocomial sources of mucorales are more frequent (such as adhesive bandages or wooden tongue depressors).
Lines 410-419: Please cite the original studies.
Line 440: Please cite the original study.
Lines 469-476: This paragraph contains redundant data with previous paragraphs (e.g. Apophysomyces has already been presented in the “soil” paragraph). It would be more interesting to present this aspect earlier. Adding a Table in the soil paragraph would be helpful to introduce these unusual species, especially Saksenaea, which, like Apophysomyces, has also been isolated following trauma with soil contact in specific areas (e.g. Middle-East).
Author Response
4.
The authors have undertaken an ambitious work by approaching numerous environmental aspects of mucorales and mucormycosis. They have identified interesting studies on ecological niches of mucorales, but the review would be improved by performing a more systematic collection of data in the different studies that were picked. The contribution of the present manuscript needs to be clarified in the paragraphs which rely on previous reviews.
The concept of this mini-review and our approach has changed radically since the original brief and submission. We have selected illustrative examples of what appear to be newly-recognised environments where cases of mucormycosis have been reported. The reference to previous, far more exhaustive systematic reviews is intentional; to set the scene for more recently described supposed environmental niches which may be from where these infections have been acquired, for example Apophysomyces infections.
We have clarified the scope. The new title describes the intended content better. We have not attempted to provide an exhaustive review of all the ecological niches but have selected illustrative examples for each environments which have not been covered in any great depth by previous reviewers.
General comments:
References should be added in the paragraphs which contain none (e.g. no reference is cited in lines 32-51, lines 218-232, lines 353-359, etc…).
Addressed
The authors have chosen to do only text description of studies. However, it would be very helpful for the reader if they could add tables to describe the reported studies. For example, for studies on soil, such a table could include geographic area, climate/soil type/cropping regimes, number of soil samples collected, mucorales species isolated, proportion when compared to other filamentous fungi, individual frequency,… It would allow a more systematic collection of data, and such tables could be useful to make further comparisons of frequencies and to identify factors associated with some mucorales genera or species.
Table 1 summarises the studies on soil samples. We thank the reviewer for this suggestion.
For some paragraphs, the text nearly only cites previous reviews. The authors could alternatively present these reviews at the beginning of the manuscript and give their arguments for not including the aspects which have already been reviewed. Otherwise, the concerned paragraphs need to be updated (see below), and references of the original studies should be cited.
Addressed in the Introduction
Specific comments:
Lines 22 and 32: Please correct “What do know so far?”, and “The present review focusses on THE what we have learnt”
Question deleted
Lines 66-68: the sentence seems contradictory. Please clarify if Lichtheimia species are xerophilic or hydrophilic.
We are not completely clear what the reviewer means with this but we have clarified the growth requirements for Lichtheimia species
Line 147: Please clarify the sentence, for example by adding “Apophysomyces cases in India » before “accounts for nearly 60% of the reported cases worldwide”. Otherwise it seems that this fungus “accounts for nearly 60% of the reported mucormycosis cases worldwide”.
This has been clarified
Lines 203-208 and lines 215-216: Please cite the original references of these studies. Further details on the 2004 tsunami would be needed here.
We have referred to our previous review where the details of these studies are given, and references. This is to avoid cluttering the present mini-review with numerous references.
Lines 209-215: This part seems not to be adequate in this paragraph, where rather context, number of cases, and/or species isolated in natural disasters are expected.
The text has been revised
Lines 275-312: Although this paragraph is entitled “Indoor air”, is seems that most data concern outdoor air. Moreover this paragraph mainly relies on a review that was published in 2009. Is there any recent data that could be added? The authors give a lot of information that does not concern mucorales, but, for example in lines 310-311, it would be interesting to have the frequencies of mucorales species. Please cite the original references of studies.
The structure has been revised. Please see our response above re referencing.
Line 402: Healthcare-associated mucormycosis have also been reviewed by Rammaert et al. in 2012. Building construction is mainly underlined by the authors in this paragraph, but other nosocomial sources of mucorales are more frequent (such as adhesive bandages or wooden tongue depressors).
Rammaeri review cited and summarised with our own commentary regarding the implication that either the environment or medical devices were contaminated . Thank you for this. Other sources discussed and cited in our 2009 review.
Lines 410-419: Please cite the original studies.
Please see our response above re referencing.
Line 440: Please cite the original study.
Please see our response above re referencing.
Lines 469-476: This paragraph contains redundant data with previous paragraphs (e.g. Apophysomyces has already been presented in the “soil” paragraph). It would be more interesting to present this aspect earlier. Adding a Table in the soil paragraph would be helpful to introduce these unusual species, especially Saksenaea, which, like Apophysomyces, has also been isolated following trauma with soil contact in specific areas (e.g. Middle-East).
The structure has been revised and a table is now provided (Table 1)